# The Impact of Digital Technologies on Company Restoration Time Following the COVID-19 Pandemic

**Giorgia Sammarco, Daniel Ruzza, Behzad Maleki Vishkaei and Pietro De Giovanni \***

Department of Business and Management, Luiss University, 00197 Rome, Italy
* Correspondence: pdegiovanni@luiss.it

**Abstract:** The global spread of COVID-19 affected societies and economies at large with simultaneous disruptions to both supply and demand. To investigate the implications of COVID-19, this study seeks to inquire into how companies can achieve resilience through performance robustness and how this affects the restoration time (i.e., the time period from the problem occurrence to the time when the company performance returns to the previous operational level). Moreover, it studies how digital technology adoption allowed firms to become more resilient during the COVID-19 outbreak by exploiting high robustness and improving restoration time. Our findings reveal that, among the several performance indicators investigated, only the robustness of Sales is of particular significance in contributing to restoration time. As regards the technologies deployed, Blockchain, 3D Printing, and Artificial Intelligence had a positive impact on firms' resilience during the outbreak.

**Keywords:** restoration time; COVID-19; resilience; digital technologies; regression analysis





## 1. Introduction

The COVID-19 pandemic has led to a tragic loss of human lives worldwide and represents a dramatic crisis in global health systems. Since the emergence of the novel coronavirus in the city of Wuhan in China in late 2019, the virus has spread to every continent, over time becoming a global pandemic [1]. However, the pandemic is far more than a health crisis, as it has severely affected societies and economies at large [2]. The economic and social disruption has affected all citizens around the world and has had wide-ranging impacts on all sectors. The consequent recession is likely to be unusual in many respects, affecting public health, equality, employment, and economic prospects.

The spread of the virus was expedited by the underlying interconnectedness and frailties of globalization and has spread along the travel patterns and trade corridors that represent the principal arteries of the global economy [3]. Growth outcomes were already weak and deteriorating at the onset of the pandemic, with a slowdown in almost all economies in 2019 and stagnating global trade [4]. Indeed, the pandemic involved simultaneous disruptions to both supply and demand in the deeply interconnected and globalized world economy [2]. To contain the spread of the virus, most countries resorted to stringent lockdown measures. For example, the Italian government declared the pandemic a national crisis in March 2020. However, these containment measures disrupted most economic activities, as workers were required to stay at home and many businesses temporarily closed, affecting international supply chains (SCs) as well as trade. On the one side, demand for many goods and services declined since households and enterprises were either physically or financially unable to maintain their spending levels. Layoffs, income losses, and economic prospects reduced household consumption and business investments. On the other side, the spread of the infection reduced labor supply and productivity, with consequent business closures, job losses, and income declines [5].

Due to the SC disruption and the economic and financial challenges caused by the COVID-19 outbreak, companies needed to preserve their sustainability performance [6] and

adapt their businesses by making significant changes. Despite the worldwide disruption, some SCs have been able to better withstand and manage the disruptive effects [4]. The most recent literature on the COVID-19 pandemic and supply-chain-related studies [7] concerns resilience strategies to deal with the pandemic impact. However, most of the papers are based on opinion and lack empirical focus and data. Moreover, the role of digital technologies, such as Artificial Intelligence (AI), Blockchain, Internet of Things (IoT), and Data Analytics, is understudied, although more recently, some authors have made some attempts in this direction. Ref. [8], for example, analyzes the digital technology effect on restoration policies and practices in the tourism sector. Ref. [9] extends the existing Technology Acceptance Model (TAM) in order to improve its utility for analyzing digital technologies in pandemic conditions. However, certain technologies—for example, Blockchain and Drones—are hardly considered. In addition, the performance indicators of resilience are not in evidence in the recent literature on this subject.

The present work seeks to review the literature and empirically investigate the importance of firms' performance robustness to enable their resilience during disruptive events. It investigates the definitions of resilience and robustness and their implications. Particularly, our work analyzes how specific assumed performance indicators—i.e., Stock Availability, Customer Service, Sales, Return on Investment, Product Quality, Market Share, and On-Time Delivery—can reduce Restoration Time, thereby enabling resilience without contributions from any other external factors. In this study, a performance indicator is considered "robust" when it has a positive effect on Restoration Time by decreasing it.

Furthermore, it has been argued that there is a significant link between innovation and knowledge preparedness. Digital technologies may have a substantial influence on firms' agility, adaptability, and alignment, and also on SC performance [10]. Therefore, this research aims to demonstrate that the adoption of certain technologies during a disruptive event, such as the COVID-19 outbreak, can have a positive impact on organizations' ability to be resilient. In order to study their impact, several digital technologies are investigated, namely, Blockchain, 3D Printing, AI, Big Data, Mobile Apps, Drones, and Social Networks. To pursue these research goals, two linear regressions were run using a sample of 425 Italian companies out of 635 worldwide respondents to the questionnaire provided. The content of this article is structured as follows. In Section 2, the existing literature is analyzed, with a specific focus on each research variable, and the theoretical hypotheses under investigation are also defined. In Section 3, the methodology is presented, including a description of the data collection and selection techniques. Section 4 is dedicated to the empirical results and the discussion and consequent managerial implications of these findings. Finally, in Section 5, research conclusions and limitations are presented.

## 2. Literature Review

The literature review is organized into two sub-sections. In the first sub-section, we look at the resilience concept, providing a definition and clarifying the distinction with similar concepts in the literature. Then, we identify the performance indicators that we used to measure resilience during the restoration time. In the second sub-section, we consider the impact of digital technologies on supply chain resilience. Their adoption reshapes the supply chain with an impact on resilience. In addition, we present the digital technologies considered in this paper. The related hypotheses for the research are indicated at the end of the appropriate sub-section or sub-sub-section (as Hx).

### 2.1. Resilience and Robustness

The COVID-19 pandemic and the consequent lockdown-enforced measures caused the disruption of the global economy and had severe consequences for companies. It has been estimated that the pandemic impacted 94% of Fortune 1000 companies [6]. SC disruption results from an unexpected event that impairs the continuity of operations and prevents companies from maintaining desired service levels, delivery quality, and delivery time [11]. In the pandemic context, three components are related to SC risks: The presence

of long-term disruptive impacts and their unpredictable scaling, the ripple effect, i.e., the concurrent disruption propagation in the SC, and the simultaneous disruptions in supply, demand, and logistics infrastructure [8]. Therefore, the severe impacts that the pandemic had on organizations worldwide demonstrated a clear need to build resilience, which is the ability of firms to restore their performance to pre-disruption levels after absorbing its effects [12]. Over the last few decades, the original concept of resilience has been developed into not only the ability to react to events but also the capacity to prepare for recovery in the short term through the effective and efficient "proactive planning of internal and external resources of the organization" [12]. Additionally, what enables a rapid financial recovery after a disruptive event is a firm's ability to have high reactivity [3]. The concept of resilience is often used interchangeably with other attributes, such as robustness [12]. However, robustness refers to a firm's ability to maintain high performance despite internal or external disruptions [13]. Therefore, while resilience accepts some performance loss in the short term by waiting for the performance to recover over time, robustness refers to the ability of firms to resist disruption and promptly minimize losses, maintaining their performance in a fairly unchanged state.

A number of indicators can be studied to analyze whether a company can survive disruptive events on its own. Performance indicators are well suited to study resilience since they respond to events over time and offer a quantitative representation of a firm's capacity to recover [14]. Previous works have used indicators such as return on investment (ROI) to conduct a resilience analysis [1,3]. Therefore, in this study, we build upon the existing literature to identify the indicators used to analyze resilience. The identified indicators are: Stock Availability, Customer Service, Sales, ROI, Product Quality, Market Share, and On-Time Delivery. They are considered robust when they can reduce Restoration Time, which represents company resilience in the case of a disruption. To our knowledge, this is the first paper that measures resilience in terms of restoration time and analyzes the firms' performance robustness to verify the existence of resilience.

### 2.1.1. Stock Availability

As a result of the outbreak, the global demand and supply of products and services experienced a dramatic change. This disruption undermined the ability of firms to guarantee the constant matching of supply and demand [14,15]. It required companies to identify and assess the most effective inventory practices to implement, such as "pre-positioning extra inventory or raising capacity prior to demand recovery" [7]. Furthermore, omnichannel strategies also assist companies by optimally managing stocks and effectively integrating them across all channels, effectively keeping stocks under control. In conclusion, adapting inventory levels and managing Stock Availability in response to a disruptive event serve to create a shield and provide an adequate degree of protection, enabling resilient SC performance [14]. Accordingly, the following hypothesis is offered:

**H1a.** *Stock Availability robustness enables companies' resilience during the COVID-19 outbreak.*

### 2.1.2. Customer Service

Customer Service plays a crucial role in increasing Product Quality and achieving a competitive advantage for firms, thereby representing a potential enabler of resilience. Moreover, Customer Service has the potential to generate new entrepreneurial opportunities, which enhances sales volume. Allowing for long-term relationships with customers, Customer Service can overcome customers' doubts and reservations about the decision-making process, and in turn, establish customer loyalty, which assets the companies can decidedly capitalize on. In the case of disruption, the ability to retain existing customers and harness favorable responses is more relevant than attracting new customers. Accordingly, the following hypothesis is offered:

**H1b.** *Customer Service robustness enables companies' resilience during the COVID-19 outbreak.*

### 2.1.3. Sales

The pandemic has had an extensive impact on Sales because of demand shocks and changes in Sales channels, with many operations shifting from physical to digital [16]. In disruptive events, the need for flexibility through adaptive selling emerges, given the scarcity of new Sales opportunities. Adaptive selling enables companies to keep pace with ever-changing demand in a volatile Sales environment [17]. Moreover, disruptions can represent a valuable source of new Sales opportunities for prospective customers. Accordingly, the following hypothesis is offered:

**H1c.** *Sales robustness enables companies' resilience during the COVID-19 outbreak.*

### 2.1.4. Return on Investment

Companies rely on economic criteria to assess their financial health and also to evaluate the economic sustainability of any process. Indeed, ROI is adopted by companies to gain valuable insights into how investments can be recovered through economic outcomes. Leveraging ROI to counteract a disruption might have widespread impacts across performance indicators. For instance, when ROI is maximized, investing additional money in quality improvement is optimal, and this, in turn, allows companies to quantitatively reduce the level of inventory [17]. Accordingly, the following hypothesis is offered:

**H1d.** *ROI robustness enables companies' resilience during the COVID-19 outbreak.*

### 2.1.5. Product Quality

The ability to deliver quality products is crucial in satisfying customers and key stakeholders. Companies need to carefully evaluate the quality of products and services delivered because it affects after-sales customer relations, an essential driver of profitability. Moreover, considering that the SC aggregates several different parties with the aim of meeting customers' needs, the relationship between customers and suppliers is based on cost as well as Product Quality, delivery, and flexibility [17]. Failure to meet these requirements will expose companies to unexpected changes in supply or demand and impair their responses [18]. Accordingly, the following hypothesis is offered:

**H1e.** *Product Quality robustness enables companies' resilience during the COVID-19 outbreak.*

### 2.1.6. Market Share

Demand fluctuations in SC planning, triggered by disruptive events, may result in a loss of Market Share [19]. In the current fast-paced, competitive environment, such fluctuations should be prevented. Indeed, several scholars have argued that a large Market Share entails greater market power, which, in turn, provides firms with the possibility of obtaining benefits, such as the ability to set higher selling prices and greater negotiating power on purchase prices with suppliers. This can result in a firm's ability to maintain high-performance standards. Accordingly, the following hypothesis is offered:

**H1f.** *Market Share robustness enables companies' resilience during the COVID-19 outbreak.*

### 2.1.7. On-Time Delivery

Finally, containment measures implemented worldwide because of the pandemic restricted people's movements and propelled an increase in online shopping and delivery services of essential and non-essential goods. This spike in demand has rendered the delivery of goods to customers more challenging [20]. Therefore, companies had to make an increased effort to demonstrate their reliability by properly and timely delivering products and services. The most affected companies are those with long SC cycles between order allocation and delivery because they are more exposed [21]. Indeed, On-Time Delivery, considered a combination of delivery date and quantity, has often been used as an indicator of resilience in SC performance [22].

The identified elements of performance support companies in proactively facing challenges caused by the pandemic without the adoption of external resources, such as digital technologies. Accordingly, the following hypothesis is offered:

**H1g.** *On-Time Delivery robustness enables companies' resilience during the COVID-19 outbreak.*

### 2.2. Resilience and Digital Technologies

SC adaptation to disruption used to be considered "an external and event-driven action to return-to-normal". However, more recently, this understanding was substituted by the concept of adaptation "as a normal existence form of SC operations" [17]. This shift was enabled by the integration of physical and digital technologies. Indeed, in a disruption context, organizations cannot only concentrate on an SC disruption orientation to develop resilience but, rather, must ensure a resource reconfiguration to mitigate the disruptive impact. To pursue the objectives of this study, the resource configuration is realized by introducing or exploiting technologies, given that they are a crucial determinant of success in the emerging digital economy [23]. Therefore, the concept of resilience should be revised due to the adoption of technology in SC Management (SCM). Indeed, the COVID-19 outbreak forced companies to seek ways to overcome the challenges posed by the increased distance to customers, such as the difficulties associated with acquiring customers' data and predicting demand, as well as the impossibility of physically reaching some customers. Hence, organizations have adopted digital technologies to mitigate the worst consequences in the aftermath of the crisis. Companies should assess the most effective way to integrate technologies into their business model for the purpose of fully exploiting the "catalyst" effect of technologies and achieving responsible digitalization. This, in turn, leads to a reshaping and restructuring of SCM and operations. Digital technologies can foster and support resilience in different ways. For instance, they can contribute to the early recognition of a disruptive event and help forecast the possible scenarios through end-to-end visibility, i.e., the ability to track components and final products along the entire chain. Additionally, digital technologies allow firms to identify intrinsic strengths and weaknesses within the organization [13] that drive performance in terms of robustness and resilience. Therefore, the extant literature identifies the importance of digital technology integration and exploitation to achieve competitive advantages in SC management and to build sustainability. To analyze the impact of technology implementation on resilience during the COVID-19 pandemic, a number of technologies can be explored, namely, Blockchain, 3D Printing, AI, Big Data, Mobile Apps, Drones, and Social Networks. Indeed, several studies have posited that these technologies have a positive impact on resilience. However, none of these studies analyzed the impact of digital technologies when resilience depends on the firms' performance robustness. This research avenue is justified by the fact that the COVID-19 pandemic was the first big disruptive event occurring after digital technologies became available to firms and adopted worldwide, ushered in with the advent of Industry 4.0.

#### 2.2.1. Blockchain

One of the most disruptive technologies is Blockchain technology [23], a ledger that exists across a peer-to-peer network whereby there are no intermediaries between the players, allowing producer organizations and suppliers to deal directly with their customers [24]. Thus, Blockchain can anonymously connect all SC parties, providing a safe environment in which all parties can securely interact in different industries and sectors [25–27]. Due to its features that maintain the confidentiality, integrity, and availability of all transactions [28], Blockchain technology has the potential to transform SCM [29]. Additionally, it enhances economic performance by improving operational efficiencies and lowers costs by easing and assisting the confluence of product flows, distribution, and information flows [28]. The extant literature argues that this technology can foster resilience [3] by avoiding or otherwise reducing organizational and network risks linked to the SC [29] through the application of a "preventive and proactive approach" [17].

### 2.2.2. 3D Printing

3D Printing is also known as "additive manufacturing," since "materials are added rather than subtracted from a larger raw material object during the manufacturing process" [30]. Initially, this technology was developed with the aim of producing prototypes, but it has evolved over the last decade, thereby extending its implementation and its impact into several new areas. Basically, 3D Printing allows for the reduction of waste, enabling the development of a circular economy [23], which consequently drives sustainable technology and a smart circular system [31]. This technology can also affect mass customization, directly involving customers in the design process with a consequent impact on the SC [32]. Additionally, 3D Printing technology allows organizations to relocate manufacturing, increasing their degree of flexibility and, in turn, improving their ability to promptly respond to changes in demand [30]. These potential impacts, along with the reduction in SC complexity and the rationalization of inventory—attributable to, for instance, potential decreases in demand for the global transportation of physical goods and inventory activities [33]–emphasize the critical role that 3D Printing can play in enabling resilience.

### 2.2.3. Artificial Intelligence (AI)

AI is a system of interconnected machines that solve a business problem, exhibiting aspects of human intelligence [34]. This technology has been exploited in a wide range of business fields. However, it has relevance in SCM because of the series of complex tasks involved [35]. AI can rapidly and carefully simplify operations through the collection and elaboration of a large amount of data. It has many applications along the SC. For instance, it can track warehouse performance in terms of inventory, or it can improve cooperation among contractors and suppliers in structuring the SC, resulting in increased connectivity, transparency, and visibility [36]. Additionally, as mentioned above, in the context of 3D Printing technology, AI can enable the development of a circular economy through its application in reverse logistics [36] using existing resources in an efficient and effective way [37]. Therefore, it is broadly argued that AI can contribute to the development of business continuity capabilities and resilience [38].

### 2.2.4. Big Data

Big Data is another notable technology that can be integrated by organizations to overcome the challenges caused by the pandemic. Big Data involves noteworthy datasets to be collected, stored, managed, and analyzed, which are handled in conformance with four qualities, namely: Volume, which refers to data size; variety, which represents data format; velocity, which refers to the rate of data production; and veracity, which stands for data reliability [3]. Big Data is particularly helpful when coupled with the concept of analytics, i.e., the ability to elaborate the dataset by acquiring information through the implementation of strategies that support managerial processes [39]. Big Data analytics supports the organization in improving productivity, competitiveness, and efficiency [40], along with decision-making capabilities—for instance, avoiding redundancy across the SC [39]. During disruptive events with consequent uncertainty regarding demand level, Big Data provides significant predictions to enhance SC flexibility in operations and environmental management [41]. Overall, Big Data analytics contributes to the successful management of all of the stages of a disruption event, through protection, mitigation, response, and recovery [39], with the result of developing resilience [40].

### 2.2.5. Mobile Apps

Over the last few decades, mobile technologies have increasingly become a fundamental factor in modern business, following the widespread use of the Internet and the subsequent advent of Social Networks. These new tools have prompted organizations to explore new opportunities, such as Mobile Apps, to provide new types of services to users, create new Sales channels [41], attract as many users as possible, and retain existing users

through better service [17]. Mobile Apps have allowed customers to not only purchase items but also to interact with the organization seeking feedback and comparing prices [42]. The popularity of mobile services has gradually grown in accordance with customers' desire to receive finished products or services in the place and at the time they desire. During the COVID-19 pandemic, the use of mobile technologies was further implemented and expanded since companies were required to develop new and innovative methods to manage their SCs. To prevent the impacts of the disruption, many organizations have provided expanded home delivery services, online Sales, and mobile services [3], thus implementing the innovative "bring-service-near-your-home" business model [28] to better coordinate with the whole SC [43].

### 2.2.6. Drones

Drones are unpiloted aircraft that can be remotely controlled or that fly autonomously using embedded systems. Although Drones were already widely used for the delivery of emergency medical products before the pandemic, today, Drones are considered one of the most disruptive instruments of SC management, directly connecting warehouses to customers [30]. Indeed, large companies are experimenting with parcel delivery using Drones. Moreover, this technology is completely sustainable since electricity is the only resource involved, and it is also safer because of the automated system. The pandemic has opened opportunities for the use of Drones or drone integration with other transportation modes, so-called "hybrid truck-Drones," to ensure on-time, contactless delivery of products.

### 2.2.7. Social Networks

The last technology considered is Social Networks. Companies leverage Social Networks to drive consumers through their online shopping experience. Another Sales channel experiencing increasing popularity is social media, some of which allows vendors to market their products through their sites. Due to the pervasive nature of social media, intelligent algorithms and systems are required to extract a large amount of the data generated to process the information, which can provide better value to both the users and the organizations involved. For instance, firms can continuously manage specific insights into customer preferences and market trends. Therefore, social media helps organizations to capture the value created by the complex impacts of human factors involved in internal SC flows [38]. They contribute to defining possible data-driven solutions in the case of a disruption [34], such as the COVID-19 outbreak. The existing literature highlights the significant contribution of digital technologies to firms' responsiveness to disruptive events. Accordingly, the following hypothesis is offered:

**H2.** *Digital technology adoption enables companies' resilience during the COVID-19 outbreak.*

### 3. Methodology

#### 3.1. Data Collection

In the aftermath of the first COVID-19 wave, for the purposes of this study, a questionnaire (see Appendix A) was drafted to collect information about how the companies were affected by this disruptive event. In September 2020, the questionnaire was submitted to a pool of eight professionals to verify its validity and completeness. The professionals read the questionnaire, suggested modifications to the questions, and advised that a general introduction of the framework be added to familiarize the respondents with the framework. Following its review, the questionnaire was thereafter submitted to 635 firms selected from Confindustria affiliates. With the aim of investigating the impact of the pandemic on Italian firms, we focused on resilience by conceptualizing it as "Restoration Time," which measures the period needed for firms to restore their performance to pre-pandemic levels. Therefore, respondents were asked how long it would take for their companies to recover the losses during the period March–June 2020. The respondents were invited to give the number of months their company requested to recover performance lost during that period.

Several components were taken into consideration as indicators of performance robustness, namely, Sales, ROI, Market Share, On-Time Delivery, Stock Availability, Customer Service, and Product Quality. Based on the discussion of these concepts in the literature review, Sales are taken to be a reflection of the amount of goods and services sold in a given time period. ROI is a measure of the profitability and efficiency of an investment. Market Share represents the percentage of an industry, or a market's total Sales, which is captured by a company over a given time period. On-Time Delivery refers to the rate of finished products and deliveries made on time. Stock Availability reflects the status of the company's finished products available to users based on how much inventory is in stock to meet customer demand. Customer Service represents the support that firms offer to their customers before and after the purchase of a product or service. Finally, Product Quality incorporates all of the features of a product or service that bear on its ability to satisfy consumer needs.

Based on these indicators of performance, the questionnaire asked how extensively each company's performance deteriorated during the COVID-19 pandemic. Thereafter, in order to investigate the impact of technology on performance, participants were asked whether certain technologies were adopted to counteract the worst impacts of the crisis. A dummy variable was used to signal their adoption, assuming a value of "0" in the case of the absence of technology and "1" when technologies—such as Blockchain, 3D Printing, AI, Big Data, Mobile Apps, Drones, and Social Networks—were implemented.

To pursue the objectives of this study, two linear regression models were run, as outlined below. For each model, the coefficients represent how the independent variables explain the dependent variable. In the first hypothesis, the linear regression equation represents the benchmark since the variables were not affected by any moderator, unlike the second hypothesis, which is affected by the adoption of digital technologies. Hence, the first regression refers to the companies' ability to restore business performance through their own business strengths, as represented by the performance indicators. In the second hypothesis, we tested whether the technologies could accelerate the recovery process and, consequently, the firms' resilience. We selected regression analysis because we had sufficient data to undertake this technique, which helped us explain whether the restoration time depends on two factors: Firms' performance robustness and the adoption of digital technologies. In fact, regression analysis is a well-fitting methodology to verify whether both the firms' performance robustness and the digital technology adoption explain the restoration time. The authors believe that these findings could not have been obtained by using other statistical methods like simple *t*-test, cluster analysis, and other statistical methods.

### 3.2. Data Description

The sample under investigation consists of 425 Italian companies out of 635 respondents to the questionnaire, which provides a comprehensive overview due to the heterogeneous features of the firms. All the details on the sample are displayed in Table 1.

This sample is primarily composed of small companies in terms of employees, in the range of <50 (62.12%), and other dimensions as well, in the ranges of 50–99 (12.47%), 100–200 (7.06%), and >200 (18.35%). Additionally, there are several company types under investigation, namely: Manufacturers (27.06%), Wholesalers (5.88%), Distributors (8.71%), Suppliers (10.82%), Retailers (14.59%), and other types (32.94%). Similarly, this sample provides a heterogeneous overview of different industries, i.e., Building (4.71%), Fashion and Luxury (9.88%), Food and Beverage (9.65%), Healthcare (4.24%), Industrial Production (16.94%), IT (5.41%), Logistics (9.88%), Service (25.41%), and other industries (13.88%). Finally, a wide range of top executives replied to the questionnaire under the positions of Analyst (2.12%), CEO (40.94%), CFO (5.65%), Director (8.24%), Manager (34.12%), and other professionals.

**Table 1.** Sample Composition.

| Employees | # | % | Company Type | # | % | Industry | # | % | Professionals | # | % |
|---|---|---|---|---|---|---|---|---|---|---|---|
| <50 | 264 | 62.12% | Manufacturer | 115 | 27.06% | Building | 20 | 4.71% | Analyst | 9 | 2.12% |
| 50–99 | 53 | 12.47% | Wholesaler | 25 | 5.88% | Fashion and Luxury | 42 | 9.88% | CEO | 174 | 40.94% |
| 100–200 | 30 | 7.06% | Distributor | 37 | 8.71% | Food and Beverage | 41 | 9.65% | CFO | 24 | 5.65% |
| >200 | 78 | 18.35% | Supplier | 46 | 10.82% | Healthcare | 18 | 4.24% | Director | 35 | 8.24% |
| | | | Retailer | 62 | 14.59% | Industrial production | 72 | 16.94% | Manager | 145 | 34.12% |
| | | | Other | 140 | 32.94% | IT | 23 | 5.41% | Others | 38 | 8.94% |
| | | | | | | Logistics | 42 | 9.88% | | | |
| | | | | | | Service | 108 | 25.41% | | | |
| | | | | | | Other | 59 | 13.88% | | | |

*3.3. Model Description*

To test the relationship between the indicators of performance robustness and resilience, we ran a linear regression model characterized by the equation noted below. Since it is necessary to verify this relationship without the contributions of any moderator, thus the ability of the organization to survive disruptive events relying on its own business strengths, this equation represents the benchmark for this study:

*Restoration Time* = $\beta_1$ + $\beta_2$*Sales* + $\beta_3$*ROI* + $\beta_4$*Product Quality* + $\beta_5$*Market Share* + $\beta_6$*Stock Availability* + $\beta_7$*Customer Service* + $\beta_8$*On-Time Delivery* + $\varepsilon$

where $\beta_1$ is the constant coefficient of the regression line, is the error term of the regression model, and the coefficients $\beta_2 \ldots \beta_8$ indicate whether Sales, ROI, Product Quality, Market Share, Stock Availability, Customer Service, and On-Time Delivery predict the Restoration Time.

According to the benchmark analysis, the restoration value is 16.57 months, implying, therefore, on average, it took 16.57 months for companies to recover from the disruptive shock. In contrast with the previous model that analyzed firms' ability to address pandemic challenges through their own performance robustness, the final model proposed seeks to analyze the correlation between the adoption, integration, and implementation of digital technologies, as well as the organizations' ability to restore their business performance at pre-pandemic levels. Therefore, digital technology plays the role of moderator in the relationship between Restoration Time and the above-mentioned performance indicators. To test this relationship, an additional linear regression model was run characterized by the following equation:

*Restoration Time* = *Benchmark Regression* + $\beta_9$*Sales*moderator* + $\beta_{10}$* *ROI*moderator* + $\beta_{11}$*Product Quality*moderator* + $\beta_{12}$* *Market Share*moderator* + $\beta_{13}$*Stock Availability*moderator* + $\beta_{14}$*Customer Service*moderator* + $\beta_{15}$*On-Time Delivery*moderator*

where, beyond considering the original regression model without moderators, namely, Benchmark Regression, the parameters $\beta_9 \ldots \beta_{15}$ indicate whether the joint effect between a moderator with Sales, ROI, Product Quality, Market Share, Stock Availability, Customer Service, and On-Time Delivery predict the Restoration Time.

**4. Empirical Results**

Regarding the first hypothesis, the results demonstrate the impact of each indicator on companies' resilience. Therefore, the results demonstrate which indicator firms can rely on to survive disruptive events. Similarly, concerning the second hypothesis, the results display the effect of technology adoption on firms' Restoration Times during the COVID-

19 outbreak. In both cases, negative coefficients refer to a reduction in the companies' Restoration Time during the COVID-19 outbreak, whereas positive coefficients involve an increase in Restoration Time.

### 4.1. Analyzing the First Hypothesis (H1)

The results of the regression analysis are shown in Table 2, indicating the value of each coefficient associated with the independent variables and the correlated *p*-values.

**Table 2.** Hypothesis 1.

| Hypothesis 1 | Intercept | Stock Availability | Customer Service | Sales | ROI | Product Quality | Market Share | On-Time Delivery |
|---|---|---|---|---|---|---|---|---|
| Coefficient | 16.57 | 0.005 | −0.023 | −0.093 | 0.025 | −0.011 | −0.011 | 0.001 |
| *p*-value | <0.0001 | 0.851 | 0.325 | <0.0001 | 0.317 | 0.728 | 0.773 | 0.979 |
| Result | Supported | Not supported | Not supported | Supported | Not supported | Not supported | Not supported | Not supported |

The research findings demonstrate that firms' Restoration Time value is 16.57 months without the adoption of any digital technology. Thus, companies would take 16.57 months to restore their business performance if the pandemic had ended the day after submitting the questionnaire reply. The results also prove that the companies' ability to quickly restore their business affairs to pre-pandemic levels, thus, be resilient, does not depend on Stock Availability (*Coefficient: 0.005 with p-value: >0.1*). This inconsistency is linked to the nature of the Stock Availability concept. Inventory management maximization is independent of COVID-19's disruption effects because companies always strive to optimize Stock Availability, regardless of the circumstances. Therefore, according to these results, companies cannot only concentrate on inventory management robustness to enable a more rapid Restoration Time. Similarly, the results regarding Customer Service reveal that the organizations' ability to maintain stability and recover their performance during the COVID-19 outbreak does not depend on Customer Service robustness (*Coefficient: −0.023 with p-value: >0.1*). This output can be explained by taking into consideration how this disruptive event affected consumers since their brand loyalty was not affected by the pandemic's consequences. Therefore, according to the results, enhancing Customer Service to increase organizational performance does not help the companies in the sample under investigation to survive this disruptive event without external support.

The research findings demonstrate that the first hypothesis is supported by the Sales variable result (*Coefficient: −0.093 with p-value: <0.0001*), which demonstrates that among all the indicators in the analysis, Sales are the only effective lever to foster resilience and restore performance levels. In fact, the coefficient shows a negative relationship between Sales and Restoration Time. Thus, this variable has a positive impact on the time companies need to restore their business. As mentioned in the literature review, companies undertook new ways to satisfy customers' needs and demands by continuing to provide products and services, despite all the problems linked to social distancing. Therefore, Restoration Time can be reduced if firms have strong Sales abilities, particularly considering the Sales growth during the COVID-19 period.

The results from this study demonstrate that resilience is not dependent on ROI during this outbreak (*Coefficient: 0.025 with p-value: >0.1*). This inconsistency is likely explained by the impact of the COVID-19 pandemic on ROI, which suffered a reduction over the long term, indicating that organizations cannot rely only on the robustness of ROI to recover their performance in the short term. Similarly, this model suggests that companies' Restoration Time does not depend on Product Quality robustness (*Coefficient: −0.011 with p-value: >0.1*). This result is probably attributable to the poor correlation between Product Quality and performance robustness during a disruptive event. It is easy to understand

that Product Quality takes second place in this case. Therefore, Product Quality cannot reduce Restoration Time.

A similar result is obtained by analyzing the firms' ability to be resilient through Market Share robustness. The research demonstrates its lack of statistical significance (*Coefficient: −0.011 with p-value: >0.1*). This result is most likely explained by the role of Market Share in performance, which is very context-specific, especially in the case of a pandemic. Finally, the regression analysis revealed that On-Time Delivery robustness does not enable resilience during the COVID-19 outbreak (*Coefficient: 0.001 with p-value: >0.1*). This outcome is linked to the significant challenges caused by the pandemic and the consequent SC disruption. Providing On-Time Delivery does not positively impact Restoration Time in the short term. For instance, during the first pandemic wave, Amazon was not able to guarantee delivery times, some of which have been even extended due to an abnormal increase of demand.

Accordingly, the following proposition is put forward:

**Proposition 1.** *During the COVID-19 outbreak, firms could rely on the robustness of Sales to achieve a high level of resilience, which is exemplified by a short Restoration Time. In contrast, robustness in Inventory Management, Customer Service, ROI, Product Quality, Market Share, and On-Time Delivery does not reduce Restoration Time, preventing firms' resilience.*

### 4.2. Analyzing the Second Hypothesis (H2)

The results of the regression analysis are summarized in Table 3, indicating the value of each moderator's impact on the following variables considering the following: * *p*-value < 0.001, ** *p*-value < 0.025, and *** *p*-value < 0.05.

**Table 3.** Hypothesis 2.

| Hypothesis 2 | Blockchain Moderator | 3D Printing Moderator | AI Moderator | Big Data Moderator | Mobile Apps Moderator | Drones Moderator | Social Networks Moderator |
|---|---|---|---|---|---|---|---|
| Coefficient × moderator | −0.056 ** | −2.583 * | −1.423 * | −0.488 | 0.374 | 1.261 * | 0.254 |
| Stock Availability × moderator | 0.178 | −0.470 * | −0.148 | 0.004 | 0.638 ** | 0.154 * | 0.387 |
| Customer Service × moderator | 0.266 ** | 1.684 ** | 0.314 ** | 0.2 | −0.107 | 0.636 * | −0.277 |
| Sales × moderator | −0.437 * | −0.409 * | −0.069 | −0.024 | −0.396 *** | −1.038 * | −0.016 |
| ROI × moderator | 0.136 | −0.170 | 0.129 | 0.015 | 0.600 *** | 0.821 * | 0.187 |
| Product Quality × moderator | 0.727 * | 2.083 * | 0.841 ** | 0.002 | −0.432 | −1.811 * | −0.319 |
| Market Share × moderator | −0.923 | −0.103 | −0.164 | −0.018 | −0.935 ** | 0 | −0.411 |
| On-Time Delivery × moderator | 0.558 ● | −0.076 | 0.482 ** | 0.362 | 0.383 | 0 | 0.187 |

* *p*-value < 0.1, ** *p*-value < 0.05, *** *p*-value < 0.01, ● not significant.

The above analysis demonstrates the company's ability to be resilient is slightly positively affected by the adoption of Blockchain technology (*Coefficient: −0.056 with p-value: <0.025*). The benefits of implementing Blockchain in companies' strategies to restore their performance to pre-pandemic levels are revealed by the negative relationship between Sales (*Coefficient: −0.437 with p-value: <0.001*) and Restoration Time and also between Market Share (*Coefficient: −0.923 with p-value: <0.05*) and Restoration Time. By using Blockchain technology, companies can unlock the full value of their datasets, activating a continuous flow of information that leads to better coordination across the SC. The benefits of exploiting the advantages of fostering more reliable data, transforming these data into a sustainable competitive advantage, are reflected in final consumers and, consequently, in Market Share and Sales, which enable companies' resilience. However, according to the results, Blockchain technology can hinder Restoration Time if organizations strive to cope with the pandemic disruption through Customer Service (*Coefficient: 0.266 with p-value: <0.025*), Product Quality (*Coefficient: 0.727 with p-value: <0.001*), and On-Time Delivery

(*Coefficient: 0.558 with p-value: <0.001*). These results are probably linked to the nature of Blockchain, which is not yet fully suitable for developing Customer Service as an enabler of resilience. Additionally, Blockchain could not leverage the benefits of Product Quality robustness, as well as On-Time Delivery, in the short run.

Furthermore, the research findings demonstrate the overall positive impact of the adoption of 3D Printing technology on Restoration Time (Coefficient: $-2.583$ with $p$-value: <0.001). Notwithstanding this result, according to the research analysis, the impact of 3D Printing is dependent upon the company's business. Indeed, the implementation of 3D Printing technologies can support organizations in the areas of inventory management (*Coefficient: −0.470 with a p-value: <0.001*) and Sales management (*Coefficient: −0.409 with a p-value: <0.001*) across the SC, fostering the process of resilience. However, managers should consider whether they are required to provide qualitatively different products (*Coefficient: 2.083 with a p-value: <0.001*) or products with increased customization (*Coefficient: 1.684 with a p-value: <0.025*) to satisfy consumers' needs. In these cases, the adoption of 3D Printing technology could have the potential to slow down the recovery process.

Moreover, the analysis shows the overall positive impact of the adoption of AI in fostering organizational recovery (*Coefficient: −1.423 with p-value: <0.001*). Although, as mentioned in the literature review, AI increases connectivity, transparency, and visibility—thereby enabling companies' flexibility—the results demonstrate positive relationships between Restoration Time and the following three variables: Customer Service (*Coefficient: 0.314 with p-value: <0.025*), Product Quality (*Coefficient: 0.841 with p-value: <0.025*), and On-Time Delivery (*Coefficient: 0.482 with p-value: <0.025*). This outcome is probably linked to consumers' expectations of quality, customization, and delivery, which have increased exponentially during the COVID-19 outbreak. Therefore, companies that decide to deviate their production processes from those standards integrating AI to satisfy new customers' needs should be careful to ensure that the recovery process is not slowed down.

On the contrary, the adoption of Big Data technology is not suitable for increasing firms' ability to restore their performance to pre-pandemic levels (Coefficient: $-0.488$ with $p$-value: >0.1). Although the existing literature highlights the importance of this digital technology, in this sample of companies, Big Data did not have a significant impact. Similarly, the results of the impact of the Mobile Apps on Restoration Time are inconsistent (Coefficient: 0.374 with $p$-value: >0.1).

Furthermore, the research findings show that firms' ability to be resilient is negatively affected by the adoption of drone technology (Coefficient: 1.261 with $p$-value: <0.001). Indeed, its negative impact is extended to Stock Availability (Coefficient: 0.154 with $p$-value: <0.001), Customer Service (Coefficient: 0.636 with $p$-value: <0.001), and ROI (Coefficient: 0.821 with $p$-value: <0.001). Although Drones are considered a disruptive response to last-mile inefficiencies (e.g., late deliveries, good damages, no time windows for deliveries), organizations still face the challenges of implementing this technology across the SC and, consequently, taking advantage of the extracted information. Therefore, companies should carefully assess this technology adoption because it can worsen the effect of inventory management, Customer Service, and ROI on Restoration Time. However, the results demonstrate a positive impact on Sales (Coefficient: $-1.038$ with $p$-value: <0.001) and Product Quality (Coefficient: $-1.811$ with $p$-value: <0.001). This outcome may depend on the business field. For instance, in the medical sector, Drones can assume a significant role in the quality of healthcare products delivery. Finally, the adoption of Social Networks is not suitable for enhancing companies' ability to recover their business performance (Coefficient: 0.254 with $p$-value: >0.1). This result probably stems from the fact that Social Networks are deeply rooted in a company's specific business. Thus, they would not constitute an added value in a disruptive event of this magnitude.

Accordingly, the following proposition can be derived:

**Proposition 2.** *During the COVID-19 outbreak, the adoption of Blockchain, 3D Printing, and AI enhances the firm's ability to quickly restore its business performance, whereas adopting Drones lead*

*to the opposite results. In contrast, Big Data, Mobile Apps, and Social Networks adoption do not impact the firms' Restoration Time.*

## 5. Conclusions and Future Extensions

### 5.1. Theoretical Implications

This study investigates various performance robustness indicators, including Stock Availability, Customer Service, Sales, ROI, Product Quality, Market Share, and On-Time Delivery, on companies' resilience during the COVID-19 outbreak to find out which of them is more effective in enabling resilience during the outbreak. Moreover, this study does the same analysis considering the adoption of different digital technologies, such as Blockchain, Big Data, AI, Drones, 3D Printing, and Social Networks.

In other words, the novelty of this research consists of demonstrating how firms' resilience can be enabled during disruptive events such as the COVID-19 outbreak through performance robustness and digital technologies. In principle, if companies have robust performance, the COVID-19 pandemic is less disruptive: When the firms' performance is robust enough to resist external shocks, the Restoration Time is low, and firms can quickly recover. In contrast, when firms' performance is not highly robust, the Restoration Time can be shortened by using digital technologies. In fact, the latter can support performance robustness, rendering firms more resilient by exploiting the information visibility, the lead time, the information accuracy, as well as the capacity to easily reach consumers. The main goal of the study was to provide significant evidence of the positive effects of certain performance indicators and technology adoption on companies' resilience, which refers to the ability to restore their business affairs to pre-pandemic levels. This study also sought to contribute to the existing literature by testing two hypotheses. Regarding the first hypothesis, the performance indicators under investigation were considered robust when they reduced Restoration Time caused by disruption without any external contribution, such as digital technology adoption. In the second hypothesis, the synergies between performance robustness and digital technologies were considered. Different from other studies, our analysis suggests measuring resilience as a proxy of the Restoration Time. Furthermore, we introduce the concept of performance robustness, which highlights the extent to which firms can resist external shocks and, consequently, be more resilient by shortening the Restoration Time. Finally, we suggest to firms which digital technologies to implement to improve the impact that the performance robustness has on the Restoration Time and, hence, fasten the resilience.

### 5.2. Practical and Managerial Implications

From a practical and managerial point of view, the research findings demonstrated that only the Sales variable, among the several performance indicators investigated, supported company resilience during the COVID-19 outbreak. Therefore, companies should concentrate on Sales to ensure business sustainability during disruptive events. Indeed, as mentioned in the literature review, organizational flexibility in the area of Sales enables companies to keep pace with ever-changing demand during a period of uncertainty.

Moreover, in relation to the second hypothesis, firms should consider the adoption of certain digital technologies, such as Blockchain, 3D Printing, and AI, to anticipate and recognize a disruptive event and especially to reduce Restoration Time. However, professionals should take into consideration the managerial implications of their choice of adoption, which are dependent upon the affected variable, as technologies may entail a worsening impact on certain indicators of performance. In particular, technology project implementation can be a disruptive influence in itself, requiring additional resources, new skillsets, and new ways of working, so these findings need to be set in the wider context of ongoing business operations in a pandemic period. Therefore, managers should attempt to strike a balance between the effects of technologies on resilience and the corollary consequences that affect indicators, such as the deterioration of Customer Service and Product Quality in the case of 3D Printing adoption. Therefore, as discussed above, to fully

exploit the "catalyst" effect of technologies, companies should assess the most effective way to integrate such technologies into their business model.

*5.3. Limitations and Future Research*

The authors recognize that this study has several limitations, not least in that it is based on a statistical analysis that provides some interesting results but which must be seen in the wider context of the operational problems encountered by companies in a pandemic situation. In addition, this analysis offers a perspective of a limited period only that covers the first wave of the COVID-19 pandemic. Furthermore, it also consists of a limited dataset that only includes Italian firms. Future research could profitably consider similar disruptive events to analyze how the dynamics of performance robustness influenced the dynamics of resilience. This may entail the analysis of a new dataset, which includes firms in a wider range of industries and countries. Other performance indicators could be investigated—for example, operating cash flow, working capital, or inventory turnover. Finally, considering the rapid pace of technological progress, other digital technologies (such as the use of Cloud computing and Robotics) may be studied to assess their impact on firms' resilience during a disruptive event, further advancing the existing literature. Digital technologies have transformed many aspects of human life in recent years, including communications, the workplace, entertainment, travel, banking, and shopping. These aspects are not captured by this research, and future developments in such areas can and should be pursued. Finally, considering the rising pace of technological progress, additional digital technologies may be examined to assess their impact on firms' resilience during a disruptive event, further advancing the existing literature [44] as well as the possible impact on the society [45]. Examples of such technologies are digital SCs and SC towers, the industrial Internet of things, cloud computing, the metaverse, smart contracts, robotics, digital twins, as well as all other industry 4.0 technologies [46].

**Author Contributions:** Conceptualization, D.R.; Formal analysis, G.S.; Resources, B.M.V.; Data curation, P.D.G. All authors have read and agreed to the published version of the manuscript.

**Funding:** This contribution is a part of a project that has received funding from the Lazio region ESF operational programme 2014–2020, Axis III- Education and Training, Investment Priority 10.ii)—Specific Objective 10.5, Pivotal Action 21—Call for proposals "Contributi per la permanenza nel mondo accademico delle eccellenze"—Project Title: "Implementazione di progetti di digitalizzazione responsabile nelle supply chain"—CUP (Unified Project Code): F86J20002710009.

**Institutional Review Board Statement:** Not applicable.

**Informed Consent Statement:** Not applicable.

**Data Availability Statement:** Not applicable.

**Conflicts of Interest:** The authors declare no conflict of interest.

## Appendix A. Questionnaire

*Appendix A.1. General Information*

1.  Indicate your company type

    a.   Manufacturer
    b.   Wholesaler
    c.   Distributor
    d.   Supplier
    e.   Retailer
    f.   Others

2.  Indicate the average number of your employees in the last two years.

    a.   <50
    b.   50–99
    c.   100–200

d.    >200
3.    Indicate the sector in which your company works.
4.    Indicate your corporate role (e.g., manager, managing director, CEO, etc.)

*Appendix A.2. Performance Indicators*

During the period March–June 2020, in which percentage did your company experience a deterioration of performance due to COVID-19?

1.    Sales (from 0% to 100%)
2.    ROI (from 0% to 100%)
3.    Product Quality (from 0% to 100%)
4.    Market Share (from 0% to 100%)
5.    Stock Availability (from 0% to 100%)
6.    On-Time Delivery (from 0% to 100%)

*Appendix A.3. Digital Technologies*

During the period March–June 2020, which real actions, best practices, and strategies have been adopted to properly respond to challenges due to COVID-19? (Multiple preferences are possible)

1.    Blockchain (Yes/No)
2.    3D Printing (Yes/No)
3.    Artificial Intelligence (Yes/No)
4.    Big Data (Yes/No)
5.    Mobile Apps (Yes/No)
6.    Drones (Yes/No)
7.    Social Networks (Yes/No)

*Appendix A.4. Restoration Time*

How many months would you need to restore your company's business affairs and volumes lost due to COVID-19 during the period March–June 2020? (Restoration Time).

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
