# Peer review of "The Impact of Digital Technologies on Company Restoration Time Following the COVID-19 Pandemic"

_sustainability, doi:10.3390/su142215266_

Round 1

Reviewer 1 Report

In our opinion, this goal was not achieved, because:

- the conclusions are superficial, the hypotheses do not take into account the numerous features of doing business;

- the results of the research are only subjective assessments of the interviewed respondents.

Therefore, I recommend changing the conclusions.

Author Response

All responses are detailed in the attached report.

Reviewer 2 Report

This research showed how firms’ resilience could be enabled during the Covid-19 outbreak. This research provided the (positive) significant impact of indicators and technologies adoption on companies’ resilience. The first hypothesis showed only the sale variable; therefore, the companies should rely on sales to ensure business sustainability. The second hypothesis showed that companies should select digital technologies to anticipate and recognize situations. This paper is vague and suffers from various limitations. Thus, this manuscript needs to be improved substantially considering the following points;

1.  In section 1, this aim, need of this research, research gap, and research question (s) need to be begun clearly.

2. Why did the authors focus on/select only stock availability, customer service, sales, return on investment, product quality, market share, and on-time delivery for analysis?

3.  Based on the impact of digital technologies, why do you not include data literacy or cyber security for analysis?

4. In section 2, this research should be incorporated into the theories.

5.   In section 3.1, the authors mentioned “…The questionnaire was submitted to a pool of professionals to verify its validity and completeness and, upon review, was thereby presented to a selection of 635 firms among the Confindustria affiliates…”

1)   What kind of characteristic professionals did you select to verify and validate?

2)   How many professionals did you use to verify and validate the questionnaire?

3)   Why did this research use a sample of 425 companies out of 635 worldwide respondents?

6.   In Table 1, in the case of the industry group, why did logistics, services, and other groups exclude any information from other columns such as employees and company type?

7.    In section 4, this research should explain why the authors need to use linear regression more than other methods.

8.  The multicollinearity and validation were considered in this research before analysis, weren’t they?

9.   The novelty of this research is not recognized in this manuscript.

10.   What are uncontrolled variables in this research?

11. In order to make the insight relationship of variables, the impact may not only consider the digital technologies but also socioeconomic and attitude. 

Author Response

(The authors gave the same response as above.)

Reviewer 3 Report

(Introduction)

1. we do not know about other research on this topic or a similar one. The author should talk about similar projects and research and also compare the results at the end.

(Literature review)

1. In the first section before starting the sub-sections, there is a need to explain the subsections of the literature in general and give explanations about the process of making sub-sections. 

2. The author did not mention how the indicators of performance robustness are achieved and found. did other publications mention them? If yes we need to know them, if not we need to know the process of finding them.

3. why are the hypothesis not related to each performance robustness indicator?? It is better to have the hypothesis for each indicator not one for all of them.

(Method)

1. In overall the method section must improve and it has to be rewritten.

2. How was the questionnaire designed? how it has been approved??

3. How were the companies participating in this research selected and how were they accessed and how did they respond to the questionnaire?? Who answered the questionnaire??

4. When did this research conduct??

5. what are the questions in the questionnaire?? How we can assess them until we can not see them??

(Analyse)

1.  Why was this research method analysis chosen? Why have not the other analysis methods been used? Is it enough to use this method to draw a conclusion? The author did not mention any of them

(Conclusion)

1. What is the novelty of this research?

2. What is the contribution of this research?

3. As I said before, what is the difference between this research and the previous one or similar ones??

Author Response

(The authors gave the same response as above.)

Reviewer 4 Report

I read the paper and I think the paper is fit with the journals aim. I have also some suggestions for the authors:

Abstract: is well written

Introduction: maybe in the last paragraph you can add also the novelty of the research.

Methodology to present the methodology not only the data collection maybe you can add in this part also the hypothesis development.

The methodology must be connected to the existing literature.

Data descriptions: what mines #

Empirical results

To test the relationship between the indicators of performance robustness and resili-
ence, we run a linear regression model characterized by the following equation. Since it is
necessary to verify this relationship without the contribution of any moderator, thus the
ability of the organization to survive at disruptive events with its own business strengths,
this equation represents the benchmark for this study:
Restoration Time = ?1 + ?2*Sales + ?3*ROI + ?4*Product Quality + ?5*Market Share +
?6*Stock Availability + ?7*Customer Service + ?8*On Time Delivery+ ε

That is a part of the methodology plead add it to the methodology.

I miss the connection to the literature and the personal opinion, pleas add it.

According to the above benchmark analysis, the restoration value is 16,57 months,
therefore on average, it took 16,57 months for companies to recover from the disruption
shock. In contrast with the previous model that analyzed the firms’ ability to face the pan-
demic challenges through their own performance robustness, the final model proposed
seeks to analyze the correlation between the adoption, integration, and implementation
of digital technologies and the organizations’ ability to restore their business performance
at pre-pandemic levels. Therefore, it has been added the technology moderator to study
the relationship between the restoration time and the above-mentioned performance in-
dicators. In order to test this relationship, an additional linear regression model was run
characterized by the following equation:
Restoration Time = Benchmark Regression + ?8* Sales*moderator + ?9* ROI*modera-
tor + ?10*Product Quality*moderator + ?11* Market Share*moderator + ?12*Stock Availabil-
ity*moderator + ?13*Customer Service*moderator + ?14*On Time Delivery*moderator

Also this part is methodology

I miss the connection to the literature and the personal opinion, pleas add it.

Also, a discussion part of the results it is welcomed.

Conclusion: to structure in the theoretical, practical and managerial implications. Limits and future research.

References: pleas revise the references according to the journal style.

Some additional paper that can help you:

Fülöp, M. T., Breaz, T. O., He, X., Ionescu, C. A., CordoÅŸ, G. S., & Stanescu, S. G. (2022). The role of universities' sustainability, teachers' wellbeing, and attitudes toward e-learning during COVID-19. Frontiers in Public Health10.

Akram, U., Fülöp, M. T., Tiron-Tudor, A., Topor, D. I., & CăpuÈ™neanu, S. (2021). Impact of digitalization on customers’ well-being in the pandemic period: challenges and opportunities for the retail industry. International Journal of Environmental Research and Public Health18(14), 7533.

Urzedo, D., Westerlaken, M., & Gabrys, J. (2022). Digitalizing forest landscape restoration: a social and political analysis of emerging technological practices. Environmental Politics, 1-26.

Author Response

(The authors gave the same response as above.)

Round 2

Reviewer 2 Report

I carefully read the new manuscript many times, but I still face vulnerability in the same comments and new comments. Therefore, it would be better if you take you effort in the next one. 

1) The paper is very interesting, but would greatly benefit from editorial/grammatical review. Many of the sentences are written in "spoken" English and not "written" English. Also, a verb tense should be reviewed for scientific writing and correct spelling for the English is required in many pages (e.g., "analize" in page 2).

2) A glossary is needed for this research with standard references such as digital technologies.
3) The
digital technologies have transformed many perspectives of human life in recent years, including communications, workplace, entertainment, travel, bank, and shopping. However, this research focused on some variables in digital technologies. Please give the reasons why do the research focus on only the variables.   

4) This research disappears the revised paper from United Nations (The Impact of Digital Technologies). It needs to be included. 

5) How many professionals did you use to verify and validate the questionnaire?

6) How do professionals verify and validate the data?

7) Modelling approach disappeared the linear regression (probability) and the definition for variables and units.

Author Response

We would like to thank the referee for his/her careful reading of the paper and constructive remarks. We describe the main points raised by the referee and how we have addressed these points below. We supply all responses to the comments below. All changes to the manuscript are highlighted in bold font.

General Comments:

I carefully read the new manuscript many times, but I still face vulnerability in the same comments and new comments. Therefore, it would be better if you take you effort in the next one.

Response: We have done our best to accommodate all the requests left by the Reviewer. 

1) The paper is very interesting, but would greatly benefit from editorial/grammatical review. Many of the sentences are written in "spoken" English and not "written" English. Also, a verb tense should be reviewed for scientific writing and correct spelling for the English is required in many pages (e.g., "analize" in page 2).

Response: We have made a further correction with a proof reader before resubmitting the manuscript. 

2) A glossary is needed for this research with standard references such as digital technologies.

Response: We have followed the standards of Sustainability. We did not find any publication in Sustainability including in a glossary. Also, the number of acronymous is low so a glossary will be not needed. 

3) The digital technologies have transformed many perspectives of human life in recent years, including communications, workplace, entertainment, travel, bank, and shopping. However, this research focused on some variables in digital technologies. Please give the reasons why do the research focus on only the variables.

Response: Indeed, this is a good argument and, as the Reviewer can imagine, doing an empirical research makes researchers stuck with the available data. Therefore, we do not have such information in our dataset. However, future research can have a look on such variables. Therefore, we reported this point in the future extensions.

4) This research disappears the revised paper from United Nations (The Impact of Digital Technologies). It needs to be included. 

Response: As we have been suggested to rework on the whole set of references, we have made it and some references have been excluded from the new batch. 

5) How many professionals did you use to verify and validate the questionnaire?

Response: All information and details are now reported in the paper.

6) How do professionals verify and validate the data?

Response: All information and details are now reported in the paper.

7) Modelling approach disappeared the linear regression (probability) and the definition for variables and units.

Response: The models have been moved to the section 3.3, as it was suggested to us in the previous round.

Reviewer 3 Report

The comments have been addressed.

Author Response

Reviewer's general comments

The comments have been addressed.

Our response: we thank very much the Reviewer for the wonderful experience we had when working in the revisions. The quality of the paper has considerably improved by implementing all the suggested comments. Thanks!

Reviewer 4 Report

Good luck!

Author Response

Reviewer's general comments

Good luck!

Our response: we thank very much the Reviewer for the wonderful experience we had when working in the revisions. The quality of the paper has considerably improved by implementing all the suggested comments. Thanks!

Round 3

Reviewer 2 Report

Thank you for your effort to review a new manuscript. It is better than a prior one. All comments and concerns were clearly described with the reasons and references. However, the paper needs to revise twofold as shown below.

1.          The grammar checkers

1.1               I carefully read your paper many times, but I still face the grammatical mistakes.

1.2               It would be great if the authors can write the full names before abbreviations such as AI and IoT.

1.3               Please check when the authors could use capital letters.

2.          Section 3.3 Model description

2.1       Why do the authors select linear regression to run model in this research? Can you give any reasons and references?

2.2       In general, when the authors used equations in the research, the explanations were required to descriptions such as definitions and units.

Author Response

We thank the Reviewer very much for the careful reading. We are respond to all comments hereby in bold font.

General comment:

Thank you for your effort to review a new manuscript. It is better than a prior one. All comments and concerns were clearly described with the reasons and references.

We are glad to see that the previous efforts have been considerably appreciated. We have done the same in this review round.

However, the paper needs to revise twofold as shown below.

  1. The grammar checkers

1.1               I carefully read your paper many times, but I still face the grammatical mistakes.

We are attaching to the editor's letter a proof reading certificate of the editor we contacted to make a proof reading for us prior the submission.

1.2               It would be great if the authors can write the full names before abbreviations such as AI and IoT.

All abbreviations have been carefully specified in this version.

1.3               Please check when the authors could use capital letters.

We have made a further check.

  1. Section 3.3 Model description

2.1       Why do the authors select linear regression to run model in this research? Can you give any reasons and references?

We have further explained the use of linear regression in this version by writing "We selected regression analysis because we had sufficient data to undertake this technique, which helped us explain whether the restoration time depends on the two factors: firms’ performance robustness and the adoption of digital technologies. In fact, regression analysis is a well-fitting methodology to verify whether both the firms’ performance robustness and the digital technology adoption explain the restoration time. The authors believe that these findings could not have been obtained by using other statistical methods like simple t-test, cluster analysis, and other statistical methods."

2.2       In general, when the authors used equations in the research, the explanations were required to descriptions such as definitions and units.

All equations are specified in the text.
